# SuperGLUE: A Stickier Benchmark for General-Purpose Language Understanding Systems

**Alex Wang**[*]
New York University

**Yada Pruksachatkun**[*]
New York University

**Nikita Nangia**[*]
New York University

**Amanpreet Singh**[*]
Facebook AI Research

**Julian Michael**
University of Washington

**Felix Hill**
DeepMind

**Omer Levy**
Facebook AI Research

**Samuel R. Bowman**
New York University

## Abstract

In the last year, new models and methods for pretraining and transfer learning have driven striking performance improvements across a range of language understanding tasks. The GLUE benchmark, introduced a little over one year ago, offers a single-number metric that summarizes progress on a diverse set of such tasks, but performance on the benchmark has recently surpassed the level of non-expert humans, suggesting limited headroom for further research. In this paper we present SuperGLUE, a new benchmark styled after GLUE with a new set of more difficult language understanding tasks, a software toolkit, and a public leaderboard. SuperGLUE is available at `super.gluebenchmark.com`.

## 1 Introduction

Recently there has been notable progress across many natural language processing (NLP) tasks, led by methods such as ELMo (Peters et al., 2018), OpenAI GPT (Radford et al., 2018), and BERT (Devlin et al., 2019). The unifying theme of these methods is that they couple self-supervised learning from massive unlabelled text corpora with effective adapting of the resulting model to target tasks. The tasks that have proven amenable to this general approach include question answering, textual entailment, and parsing, among many others (Devlin et al., 2019; Kitaev et al., 2019, i.a.).

In this context, the GLUE benchmark (Wang et al., 2019a) has become a prominent evaluation framework for research towards general-purpose language understanding technologies. GLUE is a collection of nine language understanding tasks built on existing public datasets, together with private test data, an evaluation server, a single-number target metric, and an accompanying expert-constructed diagnostic set. GLUE was designed to provide a general-purpose evaluation of language understanding that covers a range of training data volumes, task genres, and task formulations. We believe it was these aspects that made GLUE particularly appropriate for exhibiting the transfer-learning potential of approaches like OpenAI GPT and BERT.

The progress of the last twelve months has eroded headroom on the GLUE benchmark dramatically. While some tasks (Figure 1) and some linguistic phenomena (Figure 2 in Appendix B) measured in GLUE remain difficult, the current state of the art GLUE Score as of early July 2019 (88.4 from Yang et al., 2019) surpasses human performance (87.1 from Nangia and Bowman, 2019) by 1.3 points, and in fact exceeds this human performance estimate on four tasks. Consequently, while there

---

[*]Equal contribution. Correspondence: `glue-benchmark-admin@googlegroups.com`

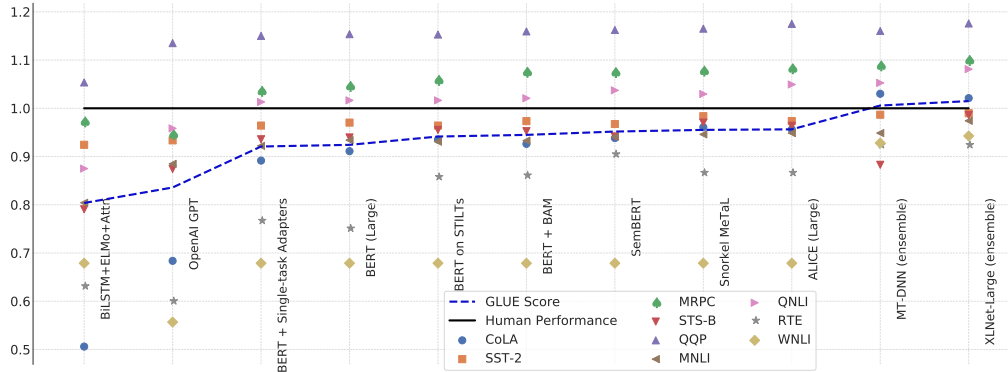

Figure 1: GLUE benchmark performance for submitted systems, rescaled to set human performance to 1.0, shown as a single number score, and broken down into the nine constituent task performances. For tasks with multiple metrics, we use an average of the metrics. More information on the tasks included in GLUE can be found in Wang et al. (2019a) and in Warstadt et al. (2019, CoLA), Socher et al. (2013, SST-2), Dolan and Brockett (2005, MRPC), Cer et al. (2017, STS-B), and Williams et al. (2018, MNLI), and Rajpurkar et al. (2016, the original data source for QNLI).

remains substantial scope for improvement towards GLUE's high-level goals, the original version of the benchmark is no longer a suitable metric for quantifying such progress.

In response, we introduce SuperGLUE, a new benchmark designed to pose a more rigorous test of language understanding. SuperGLUE has the same high-level motivation as GLUE: to provide a simple, hard-to-game measure of progress toward general-purpose language understanding technologies for English. We anticipate that significant progress on SuperGLUE should require substantive innovations in a number of core areas of machine learning, including sample-efficient, transfer, multitask, and unsupervised or self-supervised learning.

SuperGLUE follows the basic design of GLUE: It consists of a public leaderboard built around eight language understanding tasks, drawing on existing data, accompanied by a single-number performance metric, and an analysis toolkit. However, it improves upon GLUE in several ways:

**More challenging tasks:** SuperGLUE retains the two hardest tasks in GLUE. The remaining tasks were identified from those submitted to an open call for task proposals and were selected based on difficulty for current NLP approaches.

**More diverse task formats:** The task formats in GLUE are limited to sentence- and sentence-pair classification. We expand the set of task formats in SuperGLUE to include coreference resolution and question answering (QA).

**Comprehensive human baselines:** We include human performance estimates for all benchmark tasks, which verify that substantial headroom exists between a strong BERT-based baseline and human performance.

**Improved code support:** SuperGLUE is distributed with a new, modular toolkit for work on pretraining, multi-task learning, and transfer learning in NLP, built around standard tools including PyTorch (Paszke et al., 2017) and AllenNLP (Gardner et al., 2017).

**Refined usage rules:** The conditions for inclusion on the SuperGLUE leaderboard have been revamped to ensure fair competition, an informative leaderboard, and full credit assignment to data and task creators.

The SuperGLUE leaderboard, data, and software tools are available at `super.gluebenchmark.com`.

## 2 Related Work

Much work prior to GLUE demonstrated that training neural models with large amounts of available supervision can produce representations that effectively transfer to a broad range of NLP tasks

Table 1: The tasks included in SuperGLUE. *WSD* stands for word sense disambiguation, *NLI* is natural language inference, *coref.* is coreference resolution, and *QA* is question answering. For MultiRC, we list the number of total answers for 456/83/166 train/dev/test questions.

| Corpus | \|Train\| | \|Dev\| | \|Test\| | Task | Metrics | Text Sources |
|--------|--------|-------|--------|------|---------|--------------|
| BoolQ | 9427 | 3270 | 3245 | QA | acc. | Google queries, Wikipedia |
| CB | 250 | 57 | 250 | NLI | acc./F1 | various |
| COPA | 400 | 100 | 500 | QA | acc. | blogs, photography encyclopedia |
| MultiRC | 5100 | 953 | 1800 | QA | $F1_a$/EM | various |
| ReCoRD | 101k | 10k | 10k | QA | F1/EM | news (CNN, Daily Mail) |
| RTE | 2500 | 278 | 300 | NLI | acc. | news, Wikipedia |
| WiC | 6000 | 638 | 1400 | WSD | acc. | WordNet, VerbNet, Wiktionary |
| WSC | 554 | 104 | 146 | coref. | acc. | fiction books |

(Collobert and Weston, 2008; Dai and Le, 2015; Kiros et al., 2015; Hill et al., 2016; Conneau and Kiela, 2018; McCann et al., 2017; Peters et al., 2018). GLUE was presented as a formal challenge affording straightforward comparison between such task-agnostic transfer learning techniques. Other similarly-motivated benchmarks include SentEval (Conneau and Kiela, 2018), which specifically evaluates fixed-size sentence embeddings, and DecaNLP (McCann et al., 2018), which recasts a set of target tasks into a general question-answering format and prohibits task-specific parameters. In contrast, GLUE provides a lightweight classification API and no restrictions on model architecture or parameter sharing, which seems to have been well-suited to recent work in this area.

Since its release, GLUE has been used as a testbed and showcase by the developers of several influential models, including GPT (Radford et al., 2018) and BERT (Devlin et al., 2019). As shown in Figure 1, progress on GLUE since its release has been striking. On GLUE, GPT and BERT achieved scores of 72.8 and 80.2 respectively, relative to 66.5 for an ELMo-based model (Peters et al., 2018) and 63.7 for the strongest baseline with no multitask learning or pretraining above the word level. Recent models (Liu et al., 2019d; Yang et al., 2019) have clearly surpassed estimates of non-expert human performance on GLUE (Nangia and Bowman, 2019). The success of these models on GLUE has been driven by ever-increasing model capacity, compute power, and data quantity, as well as innovations in model expressivity (from recurrent to bidirectional recurrent to multi-headed transformer encoders) and degree of contextualization (from learning representation of words in isolation to using uni-directional contexts and ultimately to leveraging bidirectional contexts).

In parallel to work scaling up pretrained models, several studies have focused on complementary methods for augmenting performance of pretrained models. Phang et al. (2018) show that BERT can be improved using two-stage pretraining, i.e., fine-tuning the pretrained model on an intermediate data-rich supervised task before fine-tuning it again on a data-poor target task. Liu et al. (2019d,c) and Bach et al. (2018) get further improvements respectively via multi-task finetuning and using massive amounts of weak supervision. Clark et al. (2019b) demonstrate that knowledge distillation (Hinton et al., 2015; Furlanello et al., 2018) can lead to student networks that outperform their teachers. Overall, the quantity and quality of research contributions aimed at the challenges posed by GLUE underline the utility of this style of benchmark for machine learning researchers looking to evaluate new application-agnostic methods on language understanding.

Limits to current approaches are also apparent via the GLUE suite. Performance on the GLUE diagnostic entailment dataset, at 0.42 $R_3$, falls far below the average human performance of 0.80 $R_3$ reported in the original GLUE publication, with models performing near, or even below, chance on some linguistic phenomena (Figure 2, Appendix B). While some initially difficult categories saw gains from advances on GLUE (e.g., double negation), others remain hard (restrictivity) or even adversarial (disjunction, downward monotonicity). This suggests that even as unsupervised pretraining produces ever-better statistical summaries of text, it remains difficult to extract many details crucial to semantics without the right kind of supervision. Much recent work has made similar observations about the limitations of existing pretrained models (Jia and Liang, 2017; Naik et al., 2018; McCoy and Linzen, 2019; McCoy et al., 2019; Liu et al., 2019a,b).

Table 2: Development set examples from the tasks in SuperGLUE. **Bold** text represents part of the example format for each task. Text in *italics* is part of the model input. Underlined text is specially marked in the input. Text in a `monospaced font` represents the expected model output.

---

**BoolQ**

**Passage:** *Barq's – Barq's is an American soft drink. Its brand of root beer is notable for having caffeine. Barq's, created by Edward Barq and bottled since the turn of the 20th century, is owned by the Barq family but bottled by the Coca-Cola Company. It was known as Barq's Famous Olde Tyme Root Beer until 2012.*
**Question:** *is barq's root beer a pepsi product*     **Answer:** `No`

---

**CB**

**Text:** *B: And yet, uh, I we-, I hope to see employer based, you know, helping out. You know, child, uh, care centers at the place of employment and things like that, that will help out. A: Uh-huh. B: What do you think, do you think we are, setting a trend?*
**Hypothesis:** *they are setting a trend*     **Entailment:** `Unknown`

---

**COPA**

**Premise:** *My body cast a shadow over the grass.*     **Question:** *What's the CAUSE for this?*
**Alternative 1:** *The sun was rising.*     **Alternative 2:** *The grass was cut.*
**Correct Alternative:** `1`

---

**MultiRC**

**Paragraph:** *Susan wanted to have a birthday party. She called all of her friends. She has five friends. Her mom said that Susan can invite them all to the party. Her first friend could not go to the party because she was sick. Her second friend was going out of town. Her third friend was not so sure if her parents would let her. The fourth friend said maybe. The fifth friend could go to the party for sure. Susan was a little sad. On the day of the party, all five friends showed up. Each friend had a present for Susan. Susan was happy and sent each friend a thank you card the next week*
**Question:** *Did Susan's sick friend recover?* **Candidate answers:** *Yes, she recovered* (`T`), *No* (`F`), *Yes* (`T`), *No, she didn't recover* (`F`), *Yes, she was at Susan's party* (`T`)

---

**ReCoRD**

**Paragraph:** *(CNN) Puerto Rico on Sunday overwhelmingly voted for statehood. But Congress, the only body that can approve new states, will ultimately decide whether the status of the US commonwealth changes. Ninety-seven percent of the votes in the nonbinding referendum favored statehood, an increase over the results of a 2012 referendum, official results from the State Electorcal Commission show. It was the fifth such vote on statehood. "Today, we the people of Puerto Rico are sending a strong and clear message to the US Congress ... and to the world ... claiming our equal rights as American citizens, Puerto Rico Gov. Ricardo Rossello said in a news release. @highlight Puerto Rico voted Sunday in favor of US statehood*
**Query** For one, they can truthfully say, "Don't blame me, I didn't vote for them, " when discussing the <placeholder> presidency     **Correct Entities:** `US`

---

**RTE**

**Text:** *Dana Reeve, the widow of the actor Christopher Reeve, has died of lung cancer at age 44, according to the Christopher Reeve Foundation.*
**Hypothesis:** *Christopher Reeve had an accident.*     **Entailment:** `False`

---

**WiC**

**Context 1:** *Room and board.*     **Context 2:** *He nailed boards across the windows.*
**Sense match:** `False`

---

**WSC**

**Text:** *Mark told Pete many lies about himself, which Pete included in his book. He should have been more truthful.*     **Coreference:** `False`

---

## 3 SuperGLUE Overview

### 3.1 Design Process

The goal of SuperGLUE is to provide a simple, robust evaluation metric of any method capable of being applied to a broad range of language understanding tasks. To that end, in designing SuperGLUE, we identify the following desiderata of tasks in the benchmark:

**Task substance:** Tasks should test a system's ability to understand and reason about texts in English.

**Task difficulty:** Tasks should be beyond the scope of current state-of-the-art systems, but solvable by most college-educated English speakers. We exclude tasks that require domain-specific knowledge, e.g. medical notes or scientific papers.

**Evaluability:** Tasks must have an automatic performance metric that corresponds well to human judgments of output quality. Some text generation tasks fail to meet this criteria due to issues with automatic metrics like ROUGE and BLEU (Callison-Burch et al., 2006; Liu et al., 2016, i.a.).

**Public data:** We require that tasks have *existing* public training data in order to minimize the risks involved in newly-created datasets. We also prefer tasks for which we have access to (or could create) a test set with private labels.

**Task format:** We prefer tasks that had relatively simple input and output formats, to avoid incentivizing the users of the benchmark to create complex task-specific model architectures. Still, while GLUE is restricted to tasks involving single sentence or sentence pair inputs, for SuperGLUE we expand the scope to consider tasks with longer inputs. This yields a set of tasks that requires understanding individual tokens in context, complete sentences, inter-sentence relations, and entire paragraphs.

**License:** Task data must be available under licences that allow use and redistribution for research purposes.

To identify possible tasks for SuperGLUE, we disseminated a public call for task proposals to the NLP community, and received approximately 30 proposals. We filtered these proposals according to our criteria. Many proposals were not suitable due to licensing issues, complex formats, and insufficient headroom; we provide examples of such tasks in Appendix D. For each of the remaining tasks, we ran a BERT-based baseline and a human baseline, and filtered out tasks which were either too challenging for humans without extensive training or too easy for our machine baselines.

## 3.2 Selected Tasks

Following this process, we arrived at eight tasks to use in SuperGLUE. See Tables 1 and 2 for details and specific examples of each task.

**BoolQ** (Boolean Questions, Clark et al., 2019a) is a QA task where each example consists of a short passage and a yes/no question about the passage. The questions are provided anonymously and unsolicited by users of the Google search engine, and afterwards paired with a paragraph from a Wikipedia article containing the answer. Following the original work, we evaluate with accuracy.

**CB** (CommitmentBank, de Marneffe et al., 2019) is a corpus of short texts in which at least one sentence contains an embedded clause. Each of these embedded clauses is annotated with the degree to which it appears the person who wrote the text is *committed* to the truth of the clause. The resulting task framed as three-class textual entailment on examples that are drawn from the Wall Street Journal, fiction from the British National Corpus, and Switchboard. Each example consists of a premise containing an embedded clause and the corresponding hypothesis is the extraction of that clause. We use a subset of the data that had inter-annotator agreement above $80\%$. The data is imbalanced (relatively fewer *neutral* examples), so we evaluate using accuracy and F1, where for multi-class F1 we compute the unweighted average of the F1 per class.

**COPA** (Choice of Plausible Alternatives, Roemmele et al., 2011) is a causal reasoning task in which a system is given a premise sentence and must determine either the cause or effect of the premise from two possible choices. All examples are handcrafted and focus on topics from blogs and a photography-related encyclopedia. Following the original work, we evaluate using accuracy.

**MultiRC** (Multi-Sentence Reading Comprehension, Khashabi et al., 2018) is a QA task where each example consists of a context paragraph, a question about that paragraph, and a list of possible answers. The system must predict which answers are true and which are false. While many QA tasks exist, we use MultiRC because of a number of desirable properties: (i) each question can have multiple possible correct answers, so each question-answer pair must be evaluated independent of other pairs, (ii) the questions are designed such that answering each question requires drawing facts from multiple context sentences, and (iii) the question-answer pair format more closely matches the API of other tasks in SuperGLUE than the more popular span-extractive QA format does. The paragraphs are drawn from seven domains including news, fiction, and historical text. The evaluation metrics are F1 over all answer-options ($F1_a$) and exact match of each question's set of answers (EM).

**ReCoRD** (Reading Comprehension with Commonsense Reasoning Dataset, Zhang et al., 2018) is a multiple-choice QA task. Each example consists of a news article and a Cloze-style question about the article in which one entity is masked out. The system must predict the masked out entity from a list of possible entities in the provided passage, where the same entity may be expressed with multiple different surface forms, which are all considered correct. Articles are from CNN and Daily Mail. We evaluate with max (over all mentions) token-level F1 and exact match (EM).

**RTE** (Recognizing Textual Entailment) datasets come from a series of annual competitions on textual entailment. RTE is included in GLUE, and we use the same data and format as GLUE: We merge data from RTE1 (Dagan et al., 2006), RTE2 (Bar Haim et al., 2006), RTE3 (Giampiccolo et al., 2007), and RTE5 (Bentivogli et al., 2009). All datasets are combined and converted to two-class classification: *entailment* and *not_entailment*. Of all the GLUE tasks, RTE is among those that benefits from transfer learning the most, with performance jumping from near random-chance ($\sim$56%) at the time of GLUE's launch to 86.3% accuracy (Liu et al., 2019d; Yang et al., 2019) at the time of writing. Given the nearly eight point gap with respect to human performance, however, the task is not yet solved by machines, and we expect the remaining gap to be difficult to close.

**WiC** (Word-in-Context, Pilehvar and Camacho-Collados, 2019) is a word sense disambiguation task cast as binary classification of sentence pairs. Given two text snippets and a polysemous word that appears in both sentences, the task is to determine whether the word is used with the same sense in both sentences. Sentences are drawn from WordNet (Miller, 1995), VerbNet (Schuler, 2005), and Wiktionary. We follow the original work and evaluate using accuracy.

**WSC** (Winograd Schema Challenge, Levesque et al., 2012) is a coreference resolution task in which examples consist of a sentence with a pronoun and a list of noun phrases from the sentence. The system must determine the correct referent of the pronoun from among the provided choices. Winograd schemas are designed to require everyday knowledge and commonsense reasoning to solve.

GLUE includes a version of WSC recast as NLI, known as WNLI. Until very recently, no substantial progress had been made on WNLI, with many submissions opting to submit majority class predictions.[2] In the past few months, several works (Kocijan et al., 2019; Liu et al., 2019d) have made rapid progress via a hueristic data augmentation scheme, raising machine performance to 90.4% accuracy. Given estimated human performance of $\sim$96%, there is still a gap between machine and human performance, which we expect will be relatively difficult to close. We therefore include a version of WSC cast as binary classification, where each example consists of a sentence with a marked pronoun and noun, and the task is to determine if the pronoun refers to that noun. The training and validation examples are drawn from the original WSC data (Levesque et al., 2012), as well as those distributed by the affiliated organization *Commonsense Reasoning*.[3] The test examples are derived from fiction books and have been shared with us by the authors of the original dataset. We evaluate using accuracy.

## 3.3 Scoring

As with GLUE, we seek to give a sense of aggregate system performance over all tasks by averaging scores of all tasks. Lacking a fair criterion with which to weight the contributions of each task to the overall score, we opt for the simple approach of weighing each task equally, and for tasks with multiple metrics, first averaging those metrics to get a task score.

## 3.4 Tools for Model Analysis

**Analyzing Linguistic and World Knowledge in Models** GLUE includes an expert-constructed, diagnostic dataset that automatically tests models for a broad range of linguistic, commonsense, and world knowledge. Each example in this broad-coverage diagnostic is a sentence pair labeled with a three-way entailment relation (*entailment*, *neutral*, or *contradiction*) and tagged with labels that indicate the phenomena that characterize the relationship between the two sentences. Submissions to the GLUE leaderboard are required to include predictions from the submission's MultiNLI classifier on the diagnostic dataset, and analyses of the results were shown alongside the main leaderboard. Since this diagnostic task has proved difficult for top models, we retain it in SuperGLUE. However, since MultiNLI is not part of SuperGLUE, we collapse *contradiction* and *neutral* into a single *not_entailment* label, and request that submissions include predictions on the resulting set from the model used for the *RTE* task. We estimate human performance following the same procedure we use

for the benchmark tasks (Section C). We estimate an accuracy of 88% and a Matthew's correlation coefficient (MCC, the two-class variant of the $R_3$ metric used in GLUE) of 0.77.

**Analyzing Gender Bias in Models**    Recent work has identified the presence and amplification of many social biases in data-driven machine learning models (Lu et al., 2018; Zhao et al., 2018, i.a.). To promote the detection of such biases, we include Winogender (Rudinger et al., 2018) as an additional diagnostic dataset. Winogender is designed to measure gender bias in coreference resolution systems. We use the Diverse Natural Language Inference Collection (Poliak et al., 2018) version that casts Winogender as a textual entailment task.Each example consists of a premise sentence with a male or female pronoun and a hypothesis giving a possible antecedent of the pronoun. Examples occur in *minimal pairs*, where the only difference between an example and its pair is the gender of the pronoun in the premise. Performance on Winogender is measured with accuracy and the *gender parity score*: the percentage of minimal pairs for which the predictions are the same. A system can trivially obtain a perfect gender parity score by guessing the same class for all examples, so a high gender parity score is meaningless unless accompanied by high accuracy. We collect non-expert annotations to estimate human performance, and observe an accuracy of 99.7% and a gender parity score of 0.99.

Like any diagnostic, Winogender has limitations. It offers only positive predictive value: A poor bias score is clear evidence that a model exhibits gender bias, but a good score does not mean that the model is unbiased. More specifically, in the DNC version of the task, a low gender parity score means that a model's prediction of textual entailment can be changed with a change in pronouns, all else equal. It is plausible that there are forms of bias that are relevant to target tasks of interest, but that do not surface in this setting (Gonen and Goldberg, 2019). Also, Winogender does not cover all forms of social bias, or even all forms of gender. For instance, the version of the data used here offers no coverage of gender-neutral *they* or non-binary pronouns. Despite these limitations, we believe that Winogender's inclusion is worthwhile in providing a coarse sense of how social biases evolve with model performance and for keeping attention on the social ramifications of NLP models.

# 4   Using SuperGLUE

**Software Tools**    To facilitate using SuperGLUE, we release `jiant` (Wang et al., 2019b),[4] a modular software toolkit, built with PyTorch (Paszke et al., 2017), components from AllenNLP (Gardner et al., 2017), and the `transformers` package.[5] `jiant` implements our baselines and supports the evaluation of custom models and training methods on the benchmark tasks. The toolkit includes support for existing popular pretrained models such as OpenAI GPT and BERT, as well as support for multistage and multitask learning of the kind seen in the strongest models on GLUE.

**Eligibility**    Any system or method that can produce predictions for the SuperGLUE tasks is eligible for submission to the leaderboard, subject to the data-use and submission frequency policies stated immediately below. There are no restrictions on the type of methods that may be used, and there is no requirement that any form of parameter sharing or shared initialization be used across the tasks in the benchmark. To limit overfitting to the private test data, users are limited to a maximum of two submissions per day and six submissions per month.

**Data**    Data for the tasks are available for download through the SuperGLUE site and through a download script included with the software toolkit. Each task comes with a standardized training set, development set, and *unlabeled* test set. Submitted systems may use any public or private data when developing their systems, with a few exceptions: Systems may only use the SuperGLUE-distributed versions of the task datasets, as these use different train/validation/test splits from other public versions in some cases. Systems also may not use the unlabeled test data for the tasks in system development in any way, may not use the structured source data that was used to collect the WiC labels (sense-annotated example sentences from WordNet, VerbNet, and Wiktionary) in any way, and may not build systems that share information across separate *test* examples in any way.

To ensure reasonable credit assignment, because we build very directly on prior work, we ask the authors of submitted systems to directly name and cite the specific datasets that they use, *including the benchmark datasets*. We will enforce this as a requirement for papers to be listed on the leaderboard.

Table 3: Baseline performance on the SuperGLUE test sets and diagnostics. For CB we report accuracy and macro-average F1. For MultiRC we report F1 on all answer-options and exact match of each question's set of correct answers. $AX_b$ is the broad-coverage diagnostic task, scored using Matthews' correlation (MCC). $AX_g$ is the Winogender diagnostic, scored using accuracy and the gender parity score (GPS). All values are scaled by 100. The *Avg* column is the overall benchmark score on non-$AX_*$ tasks. The bolded numbers reflect the best machine performance on task. *MultiRC has multiple test sets released on a staggered schedule, and these results evaluate on an installation of the test set that is a subset of ours.

| Model Metrics | Avg | BoolQ Acc. | CB F1/Acc. | COPA Acc. | MultiRC F1$_a$/EM | ReCoRD F1/EM | RTE Acc. | WiC Acc. | WSC Acc. | $AX_b$ MCC | $AX_g$ GPS  Acc. |
|---|---|---|---|---|---|---|---|---|---|---|---|
| Most Frequent | 47.1 | 62.3 | 21.7/48.4 | 50.0 | 61.1 / 0.3 | 33.4/32.5 | 50.3 | 50.0 | 65.1 | 0.0 | 100.0/ 50.0 |
| CBoW | 44.3 | 62.1 | 49.0/71.2 | 51.6 | 0.0 / 0.4 | 14.0/13.6 | 49.7 | 53.0 | 65.1 | -0.4 | 100.0/ 50.0 |
| BERT | 69.0 | 77.4 | 75.7/83.6 | 70.6 | 70.0 / 24.0 | 72.0/71.3 | 71.6 | **69.5** | 64.3 | 23.0 | 97.8 / 51.7 |
| BERT++ | **71.5** | 79.0 | **84.7/90.4** | 73.8 | 70.0 / 24.1 | 72.0/71.3 | 79.0 | **69.5** | 64.3 | 38.0 | 99.4 / 51.4 |
| Outside Best | - | **80.4** | - / - | **84.4** | **70.4*/24.5*** | **74.8/73.0** | **82.7** | - | - | - | - / - |
| Human (est.) | 89.8 | 89.0 | 95.8/98.9 | 100.0 | 81.8*/51.9* | 91.7/91.3 | 93.6 | 80.0 | 100.0 | 77.0 | 99.3 / 99.7 |

# 5 Experiments

## 5.1 Baselines

**BERT**  Our main baselines are built around BERT, variants of which are among the most successful approach on GLUE at the time of writing. Specifically, we use the `bert-large-cased` variant. Following the practice recommended in Devlin et al. (2019), for each task, we use the simplest possible architecture on top of BERT. We fine-tune a copy of the pretrained BERT model separately for each task, and leave the development of multi-task learning models to future work. For training, we use the procedure specified in Devlin et al. (2019): We use Adam (Kingma and Ba, 2014) with an initial learning rate of $10^{-5}$ and fine-tune for a maximum of 10 epochs.

For classification tasks with sentence-pair inputs (BoolQ, CB, RTE, WiC), we concatenate the sentences with a [SEP] token, feed the fused input to BERT, and use a logistic regression classifier that sees the representation corresponding to [CLS]. For WiC, we also concatenate the representation of the marked word. For COPA, MultiRC, and ReCoRD, for each answer choice, we similarly concatenate the context with that answer choice and feed the resulting sequence into BERT to produce an answer representation. For COPA, we project these representations into a scalar, and take as the answer the choice with the highest associated scalar. For MultiRC, because each question can have more than one correct answer, we feed each answer representation into a logistic regression classifier. For ReCoRD, we also evaluate the probability of each candidate independent of other candidates, and take the most likely candidate as the model's prediction. For WSC, which is a span-based task, we use a model inspired by Tenney et al. (2019). Given the BERT representation for each word in the original sentence, we get span representations of the pronoun and noun phrase via a self-attention span-pooling operator (Lee et al., 2017), before feeding it into a logistic regression classifier.

**BERT++**  We also report results using BERT with additional training on related datasets before fine-tuning on the benchmark tasks, following the STILTs style of transfer learning (Phang et al., 2018). Given the productive use of MultiNLI in pretraining and intermediate fine-tuning of pretrained language models (Conneau et al., 2017; Phang et al., 2018, i.a.), for CB, RTE, and BoolQ, we use MultiNLI as a transfer task by first using the above procedure on MultiNLI. Similarly, given the similarity of COPA to SWAG (Zellers et al., 2018), we first fine-tune BERT on SWAG. These results are reported as BERT++. For all other tasks, we reuse the results of BERT fine-tuned on just that task.

**Other Baselines**  We include a baseline where for each task we simply predict the majority class,[6] as well as a bag-of-words baseline where each input is represented as an average of its tokens' GloVe word vectors (the 300D/840B release from Pennington et al., 2014). Finally, we list the best known result on each task as of May 2019, except on tasks which we recast (WSC), resplit (CB), or achieve

the best known result (WiC). The outside results for COPA, MultiRC, and RTE are from Sap et al. (2019), Trivedi et al. (2019), and Liu et al. (2019d) respectively.

**Human Performance**   Pilehvar and Camacho-Collados (2019), Khashabi et al. (2018), Nangia and Bowman (2019), and Zhang et al. (2018) respectively provide estimates for human performance on WiC, MultiRC, RTE, and ReCoRD. For the remaining tasks, including the diagnostic set, we estimate human performance by hiring crowdworker annotators through Amazon's Mechanical Turk platform to reannotate a sample of each test set. We follow a two step procedure where a crowd worker completes a short training phase before proceeding to the annotation phase, modeled after the method used by Nangia and Bowman (2019) for GLUE. See Appendix C for details.

## 5.2   Results

Table 3 shows results for all baselines. The most frequent class and CBOW baselines do not perform well overall, achieving near chance performance for several of the tasks. Using BERT increases the average SuperGLUE score by 25 points, attaining significant gains on all of the benchmark tasks, particularly MultiRC, ReCoRD, and RTE. On WSC, BERT actually performs worse than the simple baselines, likely due to the small size of the dataset and the lack of data augmentation. Using MultiNLI as an additional source of supervision for BoolQ, CB, and RTE leads to a 2-5 point improvement on all tasks. Using SWAG as a transfer task for COPA sees an 8 point improvement.

Our best baselines still lag substantially behind human performance. On average, there is a nearly 20 point gap between BERT++ and human performance. The largest gap is on WSC, with a 35 point difference between the best model and human performance. The smallest margins are on BoolQ, CB, RTE, and WiC, with gaps of around 10 points on each of these. We believe these gaps will be challenging to close: On WSC and COPA, human performance is perfect. On three other tasks, it is in the mid-to-high 90s. On the diagnostics, all models continue to lag significantly behind humans. Though all models obtain near perfect gender parity scores on Winogender, this is due to the fact that they are obtaining accuracy near that of random guessing.

## 6   Conclusion

We present SuperGLUE, a new benchmark for evaluating general-purpose language understanding systems. SuperGLUE updates the GLUE benchmark by identifying a new set of challenging NLU tasks, as measured by the difference between human and machine baselines. The set of eight tasks in our benchmark emphasizes diverse task formats and low-data training data tasks, with nearly half the tasks having fewer than 1k examples and all but one of the tasks having fewer than 10k examples.

We evaluate BERT-based baselines and find that they still lag behind humans by nearly 20 points. Given the difficulty of SuperGLUE for BERT, we expect that further progress in multi-task, transfer, and unsupervised/self-supervised learning techniques will be necessary to approach human-level performance on the benchmark. Overall, we argue that SuperGLUE offers a rich and challenging testbed for work developing new general-purpose machine learning methods for language understanding.

## 7   Acknowledgments

We thank the original authors of the included datasets in SuperGLUE for their cooperation in the creation of the benchmark, as well as those who proposed tasks and datasets that we ultimately could not include. This work was made possible in part by a donation to NYU from Eric and Wendy Schmidt made by recommendation of the Schmidt Futures program. We gratefully acknowledge the support of the NVIDIA Corporation with the donation of a Titan V GPU used at NYU for this research, and funding from DeepMind for the hosting of the benchmark platform. AW is supported by the National Science Foundation Graduate Research Fellowship Program under Grant No. DGE 1342536. Any opinions, findings, and conclusions or recommendations expressed in this material are those of the author(s) and do not necessarily reflect the views of the National Science Foundation. SB is partly supported by Samsung Advanced Institute of Technology (Next Generation Deep Learning: from Pattern Recognition to AI) and Samsung Electronics (Improving Deep Learning using Latent Structure).

## Footnotes

[2]WNLI is especially difficult due to an adversarial train/dev split: Premise sentences that appear in the training set often appear in the development set with a different hypothesis and a flipped label. If a system memorizes the training set, which was easy due to the small size of the training set, it could perform far *below* chance on the development set. We remove this adversarial design in our version of WSC by ensuring that no sentences are shared between the training, validation, and test sets.

[3]`http://commonsensereasoning.org/disambiguation.html`

[4] `https://github.com/nyu-mll/jiant`

[5] `https://github.com/huggingface/transformers`

[6]For ReCoRD, we predict the entity that has the highest F1 with the other entity options.

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
