[Supplementary Material]

Table 4: Baseline performance on the SuperGLUE development.

| Model Metrics | Avg | BoolQ Acc. | CB Acc./F1 | COPA Acc. | MultiRC $F1_a$/EM | ReCoRD F1/EM | RTE Acc. | WiC Acc. | WSC Acc. |
|---|---|---|---|---|---|---|---|---|---|
| Most Frequent Class | 47.7 | 62.2 | 50.0/22.2 | 55.0 | 59.9/ 0.8 | 32.4/31.5 | 52.7 | 50.0 | 63.5 |
| CBOW | 47.7 | 62.4 | 71.4/49.6 | 63.0 | 20.3/ 0.3 | 14.4/13.8 | 54.2 | 55.3 | 61.5 |
| BERT | 72.2 | 77.7 | 94.6/93.7 | 69.0 | 70.5/24.7 | 70.6/69.8 | 75.8 | 74.9 | 68.3 |
| BERT++ | 74.6 | 80.1 | 96.4/95.0 | 78.0 | 70.5/24.7 | 70.6/69.8 | 82.3 | 74.9 | 68.3 |

# A    Development Set Results

In Table 4, we present results of the baselines on the SuperGLUE tasks development sets.

# B    Performance on GLUE Diagnostics

Figure 2 shows the performance on the GLUE diagnostics dataset for systems submitted to the public leaderboard.

Figure 2: Performance of GLUE submissions on selected diagnostic categories, reported using the $R_3$ metric scaled up by 100, as in Wang et al. (2019a, see paper for a description of the categories). Some initially difficult categories, like double negation, saw gains from advances on GLUE, but others remain hard (restrictivity) or even adversarial (disjunction, downward monotone).

# C    Human Performance Estimation

For collecting data to establish human performance on the SuperGLUE tasks, we follow a two step procedure where we first provide some training to the crowd workers before they proceed to annotation. For both steps and all tasks, the average pay rate is $23.75/hr.[7]

In the training phase, workers are provided with instructions on the task, linked to an FAQ page, and are asked to annotate up to 30 examples from the development set. After answering each example, workers are also asked to check their work against the provided ground truth label. After the training phase is complete, we provide the qualification to work on the annotation phase to all workers who annotated a minimum of five examples, i.e. completed five HITs during training and achieved performance at, or above the median performance across all workers during training.

In the annotation phase, workers are provided with the same instructions as the training phase, and are linked to the same FAQ page. The instructions for all tasks are provided in Appendix C. For the annotation phase we randomly sample 100 examples from the task's test set, with the exception of WSC where we annotate the full test set. For each example, we collect annotations from five workers and take a majority vote to estimate human performance. For additional details, see Appendix C.3.

### C.1 Training Phase Instructions

In the training step, we provide workers with brief instructions about the training phase. An example of these instructions is given Table 5. These training instructions are the same across tasks, only the task name in the instructions is changed.

### C.2 Task Instructions

During training and annotation for each task, we provide workers with brief instructions tailored to the task. We also link workers to an FAQ page for the task. Tables 6, 7, 8, and 9, show the instructions we used for all four tasks: COPA, CommitmentBank, WSC, and BoolQ respectively. The instructions given to crowd workers for annotations on the diagnostic and bias diagnostic datasets are shown in Table 11.

We collected data to produce conservative estimates for human performance on several tasks that we did not ultimately include in our benchmark, including GAP (Webster et al., 2018), PAWS (Zhang et al., 2019), Quora Insincere Questions,[8] Ultrafine Entity Typing (Choi et al., 2018b), and Empathetic Reactions datasets (Buechel et al., 2018). The instructions we used for these tasks are shown in Tables 12, 13, 14, 15, and 16.

### C.3 Task Specific Details

For WSC and COPA we provide annotators with a two way classification problem. We then use majority vote across annotations to calculate human performance.

**CommitmentBank**  We follow the authors in providing annotators with a 7-way classification problem. We then collapse the annotations into 3 classes by using the same ranges for bucketing used by de Marneffe et al. (2019). We then use majority vote to get human performance numbers on the task.

Furthermore, for training on CommitmentBank we randomly sample examples from the low inter-annotator agreement portion of the CommitmentBank data that is not included in the benchmark version of the task. These low agreement examples are generally harder to classify since they are more ambiguous.

**Diagnostic Dataset**  Since the diagnostic dataset does not come with accompanying training data, we train our workers on examples from RTE's development set. RTE is also a textual entailment task and is the most closely related task in the main benchmark. Providing the crowd workers with training on RTE enables them to learn label definitions which should generalize to the diagnostic dataset.

**Ultrafine Entity Typing**  We cast the task into a binary classification problem to make it an easier task for non-expert crowd workers. We work in cooperation with the authors of the dataset (Choi et al., 2018b) to do this reformulation: We give workers one possible tag for a word or phrase and asked them to classify the tag as being applicable or not.

The authors used WordNet (Miller, 1995) to expand the set of labels to include synonyms and hypernyms from WordNet. They then asked five annotators to validate these tags. The tags from this validation had high agreement, and were included in the publicly available Ultrafine Entity Typing dataset,[9] This constitutes our set of positive examples. The rest of the tags from the validation procedure that are not in the public dataset constitute our negative examples.

**GAP**   For the Gendered Ambiguous Pronoun Coreference task (GAP, Webster et al., 2018), we simplified the task by providing noun phrase spans as part of the input, thus reducing the original structure prediction task to a classification task. This task was presented to crowd workers as a three way classification problem: Choose span A, B, or neither.

## D   Excluded Tasks

In this section we provide some examples of tasks that we evaluated for inclusion but ultimately could not include. We report on these excluded tasks only with the permission of their authors. We turned down many medical text datasets because they are usually only accessible with explicit permission and credentials from the data owners.

Tasks like QuAC (Choi et al., 2018a) and STREUSLE (Schneider and Smith, 2015) differed substantially from the format of other tasks in our benchmark, which we worried would incentivize users to spend significant effort on task-specific model designs, rather than focusing on general-purpose techniques. It was challenging to train annotators to do well on Quora Insincere Questions [10], Empathetic Reactions (Buechel et al., 2018), and a recast version of Ultra-Fine Entity Typing (Choi et al., 2018b, see Appendix C.3 for details), leading to low human performance. BERT achieved very high or superhuman performance on Query Well-Formedness (Faruqui and Das, 2018), PAWS (Zhang et al., 2019), Discovering Ongoing Conversations (Zanzotto and Ferrone, 2017), and GAP (Webster et al., 2018).

During the process of selecting tasks for our benchmark, we collected human performance baselines and run BERT-based machine baselines for some tasks that we ultimately excluded from our task list. We chose to exclude these tasks because our BERT baseline performs better than our human performance baseline or if the gap between human and machine performance is small.

On Quora Insincere Questions our BERT baseline outperforms our human baseline by a small margin: an F1 score of 67.2 versus 66.7 for BERT and human baselines respectively. Similarly, on the Empathetic Reactions dataset, BERT outperforms our human baseline, where BERT's predictions have a Pearson correlation of 0.45 on empathy and 0.55 on distress, compared to 0.45 and 0.35 for our human baseline. For PAWS-Wiki, we report that BERT achieves an accuracy of 91.9%, while our human baseline achieved 84% accuracy. These three tasks are excluded from the benchmark since our, admittedly conservative, human baselines are worse than machine performance. Our human performance baselines are subject to the clarity of our instructions (all instructions can be found in Appendix C), and crowd workers engagement and ability.

For the Query Well-Formedness task, the authors set an estimate human performance at 88.4% accuracy. Our BERT baseline model reaches an accuracy of 82.3%. While there is a positive gap on this task, the gap was smaller than we were were willing to tolerate. Similarly, on our recast version of the Ultrafine Entity Typing, we observe too small a gap between human (60.2 F1) and machine performance (55.0 F1). Our recasting for this task is described in Appendix C.2. On GAP, when taken as a classification problem without the related task of span selection (details in C.2), BERT performs (91.0 F1) comparably to our human baseline (94.9 F1). Given this small margin, we also exclude GAP.

On Discovering Ongoing Conversations, our BERT baseline achieves an F1 of 51.9 on a version of the task cast as sentence pair classification (given two snippets of texts from plays, determine if the second snippet is a continuation of the first). This dataset is very class imbalanced (90% negative), so we also experimented with a class-balanced version on which our BERT baselines achieves 88.4 F1. Qualitatively, we also found the task challenging for humans as there was little context for the text snippets and the examples were drawn from plays using early English. Given this fairly high machine performance and challenging nature for humans, we exclude this task from our benchmark.

*Instructions tables begin on the following page.*

Table 5: The instructions given to crowd-sourced worker describing the training phase for the Choice of Plausible Answers (COPA) task.

The New York University Center for Data Science is collecting your answers for use in research on computer understanding of English. Thank you for your help!

This project is a **training task** that needs to be completed before working on the main project on AMT named Human Performance: Plausible Answer. Once you are done with the training, please proceed to the main task! The qualification approval is not immediate but we will add you to our qualified workers list within a day.

In this training, you must answer the question on the page and then, to see how you did, click the **Check Work** button at the bottom of the page before hitting Submit. The Check Work button will reveal the true label. Please use this training and the provided answers to build an understanding of what the answers to these questions look like (the main project, Human Performance: Plausible Answer, does not have the answers on the page).

Table 6: Task-specific instructions for Choice of Plausible Alternatives (COPA). These instructions were provided during both training and annotation phases.

**Plausible Answer Instructions**

The New York University Center for Data Science is collecting your answers for use in research on computer understanding of English. Thank you for your help!

We will present you with a prompt sentence and a question. The question will either be about what caused the situation described in the prompt, or what a possible effect of that situation is. We will also give you two possible answers to this question. Your job is to decide, given the situation described in the prompt, which of the two options is a more plausible answer to the question:

In the following example, option 1. is a more plausible answer to the question about what caused the situation described in the prompt,

> *The girl received a trophy.*
> *What's the CAUSE for this?*
>> 1. *She won a spelling bee.*
>> 2. *She made a new friend.*

In the following example, option 2. is a more plausible answer the question about what happened because of the situation described in the prompt,

> *The police aimed their weapons at the fugitive.*
> *What happened as a RESULT?*
>> 1. *The fugitive fell to the ground.*
>> 2. *The fugitive dropped his gun.*

If you have any more questions, please refer to our FAQ page.

Table 7: Task-specific instructions for Commitment Bank. These instructions were provided during both training and annotation phases.

---

**Speaker Commitment Instructions**

The New York University Center for Data Science is collecting your answers for use in research on computer understanding of English. Thank you for your help!

We will present you with a prompt taken from a piece of dialogue, this could be a single sentence, a few sentences, or a short exchange between people. Your job is to figure out, based on this first prompt (on top), how certain the speaker is about the truthfulness of the second prompt (on the bottom). You can choose from a 7 point scale ranging from (1) completely certain that the second prompt is true to (7) completely certain that the second prompt is false. Here are examples for a few of the labels:

Choose 1 (certain that it is true) if the speaker from the first prompt definitely believes or knows that the second prompt is true. For example,

> *"What fun to hear Artemis laugh. She's such a serious child. I didn't know she had a sense of humor."*
> *"Artemis had a sense of humor"*

Choose 4 (not certain if it is true or false) if the speaker from the first prompt is uncertain if the second prompt is true or false. For example,

> *"Tess is committed to track. She's always trained with all her heart and soul. One can only hope that she has recovered from the flu and will cross the finish line."*
> *"Tess crossed the finish line."*

Choose 7 (certain that it is false) if the speaker from the first prompt definitely believes or knows that the second prompt is false. For example,

> *"Did you hear about Olivia's chemistry test? She studied really hard. But even after putting in all that time and energy, she didn't manage to pass the test".*
> *"Olivia passed the test."*

If you have any more questions, please refer to our FAQ page.

---

Table 8: Task-specific instructions for Winograd Schema Challenge (WSC). These instructions were provided during both training and annotation phases.

---

**Winograd Schema Instructions**

The New York University Center for Data Science is collecting your answers for use in research on computer understanding of English. Thank you for your help!

We will present you with a sentence that someone wrote, with one bolded pronoun. We will then ask if you if the pronoun refers to a specific word or phrase in the sentence. Your job is to figure out, based on the sentence, if the bolded pronoun refers to this selected word or phrase:

Choose Yes if the pronoun refers to the selected word or phrase. For example,

> *"I put the cake away in the refrigerator. It has a lot of butter in it."*
> *Does **It** in "It has a lot" refer to **cake**?*

Choose No if the pronoun does not refer to the selected word or phrase. For example,

> *"The large ball crashed right through the table because it was made of styrofoam."*
> *Does **it** in "it was made" refer to **ball**?*

If you have any more questions, please refer to our FAQ page.

---

Table 9: Task-specific instructions for BoolQ (continued in Table 10). These instructions were provided during both training and annotation phases.

---

**Question-Answering Instructions**

The New York University Center for Data Science is collecting your answers for use in research on computer understanding of English. Thank you for your help!

We will present you with a passage taken from a Wikipedia article and a relevant question. Your job is to decide, given the information provided in the passage, if the answer to the question is Yes or No. For example,

**In the following examples the correct answer is Yes,**

> *The thirteenth season of Criminal Minds was ordered on April 7, 2017, by CBS with an order of 22 episodes. The season premiered on September 27, 2017 in a new time slot at 10:00PM on Wednesday when it had previously been at 9:00PM on Wednesday since its inception. The season concluded on April 18, 2018 with a two-part season finale.*
> *will there be a 13th season of criminal minds?*
> (In the above example, the first line of the passage says that the 13th season of the show was ordered.)
>
> *As of 8 August 2016, the FDA extended its regulatory power to include e-cigarettes. Under this ruling the FDA will evaluate certain issues, including ingredients, product features and health risks, as well their appeal to minors and non-users. The FDA rule also bans access to minors. A photo ID is required to buy e-cigarettes, and their sale in all-ages vending machines is not permitted. The FDA in September 2016 has sent warning letters for unlawful underage sales to online retailers and retailers of e-cigarettes.*
> *is vaping illegal if you are under 18?*
> (In the above example, the passage states that the "FDA rule also bans access to minors." The question uses the word "vaping," which is a synonym for e-cigarttes.)

**In the following examples the correct answer is No,**

> *Badgers are short-legged omnivores in the family Mustelidae, which also includes the otters, polecats, weasels, and wolverines. They belong to the caniform suborder of carnivoran mammals. The 11 species of badgers are grouped in three subfamilies: Melinae (Eurasian badgers), Mellivorinae (the honey badger or ratel), and Taxideinae (the American badger). The Asiatic stink badgers of the genus Mydaus were formerly included within Melinae (and thus Mustelidae), but recent genetic evidence indicates these are actually members of the skunk family, placing them in the taxonomic family Mephitidae.*
> *is a wolverine the same as a badger?*
> (In the above example, the passage says that badgers and wolverines are in the same family, Mustelidae, which does not mean they are the same animal.)

---

Table 10: Continuation from Table 9 of task-specific instructions for BoolQ. These instructions were provided during both training and annotation phases.

---

*More famously, Harley-Davidson attempted to register as a trademark the distinctive "chug" of a Harley-Davidson motorcycle engine. On February 1, 1994, the company filed its application with the following description: "The mark consists of the exhaust sound of applicant's motorcycles, produced by V-twin, common crankpin motorcycle engines when the goods are in use." Nine of Harley-Davidson's competitors filed oppositions against the application, arguing that cruiser-style motorcycles of various brands use the same crankpin V-twin engine which produces the same sound. After six years of litigation, with no end in sight, in early 2000, Harley-Davidson withdrew their application.*

*does harley davidson have a patent on their sound?*

(In the above example, the passage states that Harley-Davidson applied for a patent but then withdrew, so they do not have a patent on the sound.)

If you have any more questions, please refer to our FAQ page.

---

Table 11: Task-specific instructions for the diagnostic and the bias diagnostic datasets. These instructions were provided during both training and annotation phases.

**Textual Entailment Instructions**

The New York University Center for Data Science is collecting your answers for use in research on computer understanding of English. Thank you for your help!

We will present you with a prompt taken from an article someone wrote. Your job is to figure out, based on this correct prompt (the first prompt, on top), if another prompt (the second prompt, on bottom) is also necessarily true:

Choose True if the event or situation described by the first prompt definitely implies that the second prompt, on bottom, must also be true. For example,

- *"Murphy recently decided to move to London."*
  *"Murphy recently decided to move to England."*
  (The above example is True because London is in England and therefore prompt 2 is clearly implied by prompt 1.)

- *"Russian cosmonaut Valery Polyakov set the record for the longest continuous amount of time spent in space, a staggering 438 days, between 1994 and 1995."*
  *"Russians hold record for longest stay in space."*
  (The above example is True because the information in the second prompt is contained in the first prompt: Valery is Russian and she set the record for longest stay in space.)

- *"She does not disagree with her brother's opinion, but she believes he's too aggresive in his defense"*
  *"She agrees with her brother's opinion, but she believes he's too aggresive in his defense"*
  (The above example is True because the second prompt is an exact paraphrase of the first prompt, with exactly the same meaning.)

Choose False if the event or situation described with the first prompt on top does not necessarily imply that this second prompt must also be true. For example,

- *"This method was developed at Columbia and applied to data processing at CERN."*
  *"This method was developed at Columbia and applied to data processing at CERN with limited success."*
  (The above example is False because the second prompt is introducing new information not implied in the first prompt: The first prompt does not give us any knowledge of how succesful the application of the method at CERN was.)

- *"This building is very tall."*
  *"This is the tallest building in New York."*
  (The above example is False because a building being tall does not mean it must be the tallest building, nor that it is in New York.)

- *"Hours earlier, Yasser Arafat called for an end to attacks against Israeli civilians in the two weeks before Israeli elections."*
  *"Arafat condemned suicide bomb attacks inside Israel."*
  (The above example is False because from the first prompt we only know that Arafat called for an end to attacks against Israeli citizens, we do not know what kind of attacks he may have been condemning.)

You do not have to worry about whether the writing style is maintained between the two prompts.

If you have any more questions, please refer to our FAQ page.

Table 12: Task-specific instructions for the Gendered Ambiguous Pronoun Coreference (GAP) task. These instructions were provided during both training and annotation phases.

---

**GAP Instructions**

The New York University Center for Data Science is collecting your answers for use in research on computer understanding of English. Thank you for your help!

We will present you with an extract from a Wikipedia article, with one bolded pronoun. We will also give you two names from the text that this pronoun could refer to. Your job is to figure out, based on the extract, if the pronoun refers to option A, options B, or neither:

Choose A if the pronoun refers to option A. For example,

> *"In 2010 Ella Kabambe was not the official Miss Malawi; this was Faith Chibale, but Kabambe represented the country in the Miss World pageant. At the 2012 Miss World, Susan Mtegha pushed Miss New Zealand, Collette Lochore, during the opening headshot of the pageant, claiming that Miss New Zealand was in her space."*
>
> *Does **her** refer to option A or B below?*

- A  *Susan Mtegha*
- B  *Collette Lochore*
- C  *Neither*

Choose B if the pronoun refers to option B. For example,

> *"In 1650 he started his career as advisor in the ministerium of finances in Den Haag. After he became a minister he went back to Amsterdam, and took place as a sort of chairing mayor of this city. After the death of his brother Cornelis, De Graeff became the strong leader of the republicans. He held this position until the rampjaar."*
>
> *Does **He** refer to option A or B below?*

- A  *Cornelis*
- B  *De Graeff*
- C  *Neither*

Choose C if the pronoun refers to neither option. For example,

> *"Reb Chaim Yaakov's wife is the sister of Rabbi Moishe Sternbuch, as is the wife of Rabbi Meshulam Dovid Soloveitchik, making the two Rabbis his uncles. Reb Asher's brother Rabbi Shlomo Arieli is the author of a critical edition of the novallae of Rabbi Akiva Eiger. Before his marriage, Rabbi Arieli studied in the Ponevezh Yeshiva headed by Rabbi Shmuel Rozovsky, and he later studied under his father-in-law in the Mirrer Yeshiva."*
>
> *Does **his** refer to option A or B below?*

- A  *Reb Asher*
- B  *Akiva Eiger*
- C  *Neither*

If you have any more questions, please refer to our FAQ page.

---

Table 13: Task-specific instructions for the Paraphrase Adversaries from Word Scrambling (PAWS) task. These instructions were provided during both training and annotation phases.

**Paraphrase Detection Instructions**

The New York University Center for Data Science is collecting your answers for use in research on computer understanding of English. Thank you for your help!

We will present you with two similar sentences taken from Wikipedia articles. Your job is to figure out if these two sentences are paraphrases of each other, and convey exactly the same meaning:

Choose Yes if the sentences are paraphrases and have the exact same meaning. For example,

> *"Hastings Ndlovu was buried with Hector Pieterson at Avalon Cemetery in Johannesburg."*
> *"Hastings Ndlovu , together with Hector Pieterson , was buried at the Avalon cemetery in Johannesburg ."*
>
> *"The complex of the Trabzon World Trade Center is close to Trabzon Airport ."*
> *"The complex of World Trade Center Trabzon is situated close to Trabzon Airport ."*

Choose No if the two sentences are not exact paraphrases and mean different things. For example,

> *"She was only a few months in French service when she met some British frigates in 1809 ."*
> *"She was only in British service for a few months , when in 1809 , she encountered some French frigates ."*
>
> *"This work caused him to trigger important reflections on the practices of molecular genetics and genomics at a time when this was not considered ethical ."*
> *"This work led him to trigger ethical reflections on the practices of molecular genetics and genomics at a time when this was not considered important ."*

If you have any more questions, please refer to our FAQ page.

Table 14: Task-specific instructions for the Quora Insincere Questions task. These instructions were provided during both training and annotation phases.

**Insincere Questions Instructions**

The New York University Center for Data Science is collecting your answers for use in research on computer understanding of English. Thank you for your help!

We will present you with a question that someone posted on Quora. Your job is to figure out whether or not this is a sincere question. An insincere question is defined as a question intended to make a statement rather than look for helpful answers. Some characteristics that can signify that a question is insincere:

- Has a non-neutral tone
  - Has an exaggerated tone to underscore a point about a group of people
  - Is rhetorical and meant to imply a statement about a group of people
- Is disparaging or inflammatory
  - Suggests a discriminatory idea against a protected class of people, or seeks confirmation of a stereotype
  - Makes disparaging attacks/insults against a specific person or group of people
  - Based on an outlandish premise about a group of people
  - Disparages against a characteristic that is not fixable and not measurable
- Isn't grounded in reality
  - Based on false information, or contains absurd assumptions
  - Uses sexual content (incest, bestiality, pedophilia) for shock value, and not to seek genuine answers

Please note that there are far fewer insincere questions than there are sincere questions! So you should expect to label most questions as sincere.

**Examples,**

Choose Sincere if you believe the person asking the question was genuinely seeking an answer from the forum. For example,

> *"How do DNA and RNA compare and contrast?"*
> *"Are there any sports that you don't like?"*
> *"What is the main purpose of penance?"*

Choose Insincere if you believe the person asking the question was not really seeking an answer but was being inflammatory, extremely rhetorical, or absurd. For example,

> *"How do I sell Pakistan? I need lots of money so I decided to sell Pakistan any one wanna buy?"*
> *"If Hispanics are so proud of their countries, why do they move out?"*
> *"Why Chinese people are always not welcome in all countries?"*

If you have any more questions, please refer to our FAQ page.

Table 15: Task-specific instructions for the Ultrafine Entity Typing task. These instructions were provided during both training and annotation phases.

---

**Entity Typing Instructions**

The New York University Center for Data Science is collecting your answers for use in research on computer understanding of English. Thank you for your help!

We will provide you with a sentence with on bolded word or phrase. We will also give you a possible tag for this bolded word or phrase. Your job is to decide, in the context of the sentence, if this tag is correct and applicable to the bolded word or phrase:

Choose Yes if the tag is applicable and accurately describes the selected word or phrase. For example,

> *"Spain was the gold line." **It** started out with zero gold in 1937, and by 1945 it had 65.5 tons.*
> *Tag: nation*

Choose No if the tag is not applicable and does not describes the selected word or phrase. For example,

> ***Iraqi museum workers** are starting to assess the damage to Iraq's history.*
> *Tag: organism*

If you have any more questions, please refer to our FAQ page.

---

Table 16: Task-specific instructions for the Empathetic Reaction task. These instructions were provided during both training and annotation phases.

---

**Empathy and Distress Analysis Instructions**

The New York University Center for Data Science is collecting your answers for use in research on computer understanding of English. Thank you for your help!

We will present you with a message someone wrote after reading an article. Your job is to figure out, based on this message, how disressed and empathetic the author was feeling. Empathy is defined as feeling warm, tender, sympathetic, moved, or compassionate. Distressed is defined as feeling worried, upset, troubled, perturbed, grieved, distrubed, or alarmed.

**Examples,**
The author of the following message was not feeling empathetic at all with an empathy score of 1, and was very distressed with a distress score of 7,

> *"I really hate ISIS. They continue to be the stain on society by committing atrocities condemned by every nation in the world. They must be stopped at all costs and they must be destroyed so that they wont hurt another soul. These poor people who are trying to survive get killed, imprisoned, or brainwashed into joining and there seems to be no way to stop them."*

The author of the following message is feeling very empathetic with an empathy score of 7 and also very distressed with a distress score of 7,

> *"All of you know that I love birds. This article was hard for me to read because of that. Wind turbines are killing a lot of birds, including eagles. It's really very sad. It makes me feel awful. I am all for wind turbines and renewable sources of energy because of global warming and coal, but this is awful. I don't want these poor birds to die like this. Read this article and you'll see why."*

The author of the following message is feeling moderately empathetic with an empathy score of 4 and moderately distressed with a distress score of 4,

> *"I just read an article about wild fires sending a smokey haze across the state near the Appalachian mountains. Can you imagine how big the fire must be to spread so far and wide? And the people in the area obviously suffer the most. What if you have asthma or some other condition that restricts your breathing?"*

The author of the following message is feeling very empathetic with an empathy score of 7 and mildly distressed with a distress score of 2,

> *"This is a very sad article. Being of of the first female fighter pilots must have given her and her family great honor. I think that there should be more training for all pilots who deal in these acrobatic flying routines. I also think that women have just as much of a right to become a fighter pilot as men."*

If you have any more questions, please refer to our FAQ page.

---

## Footnotes

[7]This estimate is taken from `https://turkerview.com`.

[8]`https://www.kaggle.com/c/quora-insincere-questions-classification/data`

[9]`https://homes.cs.washington.edu/~eunsol/open_entity.html`

[10]`https://www.kaggle.com/c/quora-insincere-questions-classification/data`