[Reviews · NeurIPS 2019]

Reviewer 1



This paper establishes a new set of benchmark datasets to evaluate "general-purpose language understanding". This benchmark contains six tasks: one general-purpose NLI (RTE), one NLI focused on embedded clauses and beliefs (CB), one causality (COPA), one QA (MultiRC), one WSD (WiC), and one coref (WSC). These tasks were selected from a large number with the filtering criterion based on effectiveness of curent SOTA models such as BERT. Building systems that do well on these tasks should help advance the state-of-the-art in language understanding of this type. As the authors say, the GLUE benchmark has been very helpful as an evaluation testbed for pre-training methods such as ELMo and BERT, multi-task learning methodology, transfer learning, and more. But as performance gets higher, the shortcomings of that benchmark become more apparent, so the creation of a new set of benchmark tasks will no doubt spur more research. I'll discuss some pros and cons of this approach: Pros: (1) The dataset pulls in text from a variety of domains (Table 1), with a range of NLP problem formats and phenomena tested. (2) This dataset has fewer benchmarks than GLUE. I view this as a pro: it means that things won't be so NLI-focused by virtue of having several simliar benchmarks, and hopefully means that researchers will do more in-depth analysis on each dataset, rather than simply reporting the average (6 being a somewhat manageable number to deal with). (3) As I mentioned above, the authors are very forthcoming about the limits of their datasets (biases, focus on standard written English, etc.) which I think is a good trend. Cons: (1) The format is less uniform than GLUE. This is probably inevitable for scaling to harder problems, but may discourage some folks who were attracted to the highly uniform framework of GLUE. Neutral: (1) The datasets are less large-scale. This will probably make pre-trained approaches even more critical for high performance here, potentially reducing the scope of types of work that are evaluated here. However, there may be a lower computational barrier to entry, which may democratize research on this benchmark. ----------- Overall, this is a resource that I believe will be quite useful for a lot of authors. Its construction is not "original" as such but I believe the authors are leading by example in terms of methodology for creating task suites, which will become more important as NLP moves beyond single i.i.d. train-test splits. The work done so far is of high quality so I think we can expect that the framework code will prove to be of the same standard. This paper is very clearly written and was a pleasure to read. ============== Thanks for the author response. I had a high opinion of this work before and the authors have provided some sensible discussion of the points raised by the reviewers. I am still highly in favor of accepting this work.

Reviewer 2



# Originality The paper clearly explains the previous work. It builds upon the success of GLUE, and combines it with (1) new desiderata for task selection, (2) new governance, and (3) human baseline evaluations. # Quality The key claim in the paper was that SOTA LMs perform much worse on SuperGLUE than humans do. This claim was well supported by the experimental results in the paper. # Clarity The authors clearly explained how they evaluated the humans and the models to form their baseline, as well as the criteria used in selecting the baseline. # Significance This is the most difficult and comprehensive benchmark that I have seen for LMs, and I expect that it will help drive further research progress for the community.

Reviewer 3



It is overall a good quality paper. However, a few details could be improved. 1. Can the authors assign a name to their metric? It would help others to adopt it more easily. 2. Can the authors better categorize the tasks according to different aspects that the tasks can reveal about a system’s language understanding ability? Basically, is there a deeper motivation other than the chosen tasks are “more difficult”? 3. In terms of the reflection on a system’s language understanding ability, different tasks will have overlaps. Have the authors considered such bias in the overall metric? 4. Could the authors provide more dataset stats on WSC? The original test set is sometimes considered to be too small. How large is the test set size here? ------------------------------- I thank the authors for providing such detailed responses, which have addressed all my concerns. Thus, I am increasing the score to 8.

[Author Response · NeurIPS 2019]

We thank all the reviewers for their time and comments. As the reviewers have stated, our main contribution is providing a new benchmark for evaluating general-purpose NLU systems, which is necessary given the saturation of the GLUE benchmark. Our work builds directly on GLUE and maintains the same general structure. Given the successes of GLUE, this decision seems prudent, but we also revise the tasks and rules to address weaknesses of GLUE. Our new benchmark has a stronger focus on low-data tasks, and the datasets are more diverse and less NLI-focused. We are careful to include tasks that are challenging for machines and solvable by humans, and we provide baseline performance for both.

Our benchmark does have a less uniform API than GLUE, but we view this as both a pro and a con. With GLUE, all tasks were sentence or sentence pair classification. In early experiments exploring potential benchmark tasks, we found that BERT generally performs well on existing datasets for this type of task, likely because the pretraining procedure explicitly accounts for this format (e.g., via the segment IDs). Also, we found many tasks we wanted to include but would have required awkward format changes to fit the GLUE classification format (e.g. QA). Thus, we believe expanding the range of task formats is beneficial overall. Another advantage of the new formats is that they incentivize researchers to develop better methods for *adapting* pretrained models to relatively complex task formats, instead of only designing better pretraining objectives.

Regarding the increased focus on low-data tasks, we agree that the smaller dataset sizes increases the likelihood that transfer learning from large corpora will be important for competitive performance, as seems to be the case with GLUE. However, we suspect that there is much more research to be done on how best to realize this transfer, particularly with only a small number of samples from which to adapt. We observe that the GLUE tasks where humans still substantially outperform models are tasks with the least amounts of data. On the large-data tasks, pretrained models seem to effectively leverage the volume of data to adapt to the task, leading to super- or near-human performance. We thus feel it is timely to place the emphasis on small-data tasks.

In terms of measuring specific linguistic capabilities of systems, we provide a diagnostic dataset (AX) aimed to give users a focused analysis of their systems' language understand abilities. Each example in the diagnostic has expert labels of what types of natural language phenomena are present. Solving an example is evidence that a model has grasped the phenomena in that example. Unlike the main tasks in GLUE and SuperGLUE, the diagnostic dataset was collected from a phenomena-based distribution, and not from a more "natural" data-driven distribution.

In selecting the benchmark tasks, we did not try to first identify a set of NLU skills we thought models should have, and then set out to find tasks that test those skills. We believe that such an approach would not likely yield clear conclusions, as there is no standard list of NLU skills in NLP that we could draw on and most tasks require combining multiple NLU skills to solve (confounding conclusions we could draw about whether or not a system had learned a skill). Instead, we sought to maximize task difficulty and diversity (including diversity of some broad notion of what each task tested) within the space of existing, vetted tasks. However, we do believe that several of the tasks in our benchmark are clear in what knowledge or broad skill is being tested for. For example,

- WSC is a coreference task but is designed to require commonsense reasoning to solve.
- COPA explicitly tests systems' causal reasoning ability (somewhat related to commonsense reasoning).
- CommitmentBank tests whether sentences' truth conditions apply to embedded clauses (related to veridicality).
- MultiRC is designed to require gathering and synthesizing facts from multiple text segments.
- WiC tests whether models understand how the meaning of polysemous words vary according to context.
- RTE requires grasping a broad range of abilities. This is typical of NLP benchmark tasks (e.g. SQuAD), and we believe is especially appropriate for a benchmark for general-purpose language understanding.

Regarding bias in the type of NLU skills our benchmark tests for, some of the tasks do require the same high-level abilities to solve. However, even if the ability is the same at a high-level, the specifics of the execution and the contexts in which that ability must be applied will differ across tasks. Therefore, skill overlap across tasks is useful because we want to test whether systems can perform these high-level abilities despite surface variation between tasks. We were hesitant to add more tasks when we already had a similar task, e.g. the very many different QA datasets that are currently used. Furthermore, all of the tasks are fairly mainstream, challenging NLP tasks, which we take to mean that solving these tasks requires interesting skills outside the scope of current approaches.

The average WSC sentence length is 25 words. The test set is 146 examples (Table 1), around the same size as the version in GLUE (WNLI). Despite the small dataset, the perfect human performance on WSC suggests that it represents a real, if challenging, language understanding task, and therefore worthwhile to include. Since our NeurIPS submission, systems have made substantial progress on WNLI, indicating that the dataset is not too small to learn the task.

Finally, we agree that a short name will help and we will add one to the camera-ready version! We thank the reviewers again for their time and thoughtful feedback.

[Meta-Review · NeurIPS 2019]

This paper presents SuperGLUE a new benchmark containing a number of natural language understanding tasks. SuperGLUE is more challenging than the widely used benchmark GLUE. Pros • The paper is very well written. • The work is technically sound and very solid. • The work is original. • The work is likely to make big impact on NLP research. • Related work is described in details. Cons • There might be several shortcomings with the datasets and tasks. The reviewers liked the paper and made comments on the author response.